# Mining Key Drought-Resistant Genes of Upland Cotton Based on RNA-Seq and WGCNA Analysis

**DOI:** 10.3390/plants14101407

**Published:** 2025-05-08

**Authors:** Hu Zhang, Wen Zhang, Yu Tang, Yuantao Guo, Jinsheng Wang, Wenju Gao, Qingtao Zeng, Quanjia Chen, Qin Chen

**Affiliations:** 1Xinjiang Key Laboratory of Crop Biological Breeding, College of Agriculture, Xinjiang Agricultural University, Urumqi 830052, China; 18140750187@163.com (H.Z.); tang230220@163.com (Y.T.); 18781575186@163.com (Y.G.); 15999149052@163.com (J.W.); gaowenju120@163.com (W.G.); chqjia@126.com (Q.C.); 2The 7th Division of Agricultural Sciences Institute, Xinjiang Production and Construction Corps, Kuitun 833200, China; zhangwenshzu@163.com (W.Z.); zy59720@163.com (Q.Z.)

**Keywords:** drought stress, *Gossypium hirsutum*, weighted gene co-expression network analysis, transcriptome, WRKY transcription factor

## Abstract

Cotton, as a globally important fiber crop, is significantly affected by drought stress during production. This study uses the drought-resistant variety Jin and the drought-sensitive variety TM-1 as test materials. Through multi-period drought stress treatments at 0 d, 7 d, 10 d, 15 d, and 25 d, combined with dynamic monitoring of physiological indicators, RNA sequencing, and weighted gene co-expression network analysis, the molecular mechanism of cotton drought resistance is systematically analyzed. Dynamic monitoring of physiological indicators showed that Jin significantly accumulated proline, maintained superoxide dismutase activity, reduced malondialdehyde accumulation, and delayed chlorophyll degradation. Transcriptome analysis revealed that Jin specifically activated 8544 differentially expressed genes after stress, which were significantly enriched in lipid metabolism (α-linolenic acid, ether lipids) and secondary metabolic pathways. Weighted gene co-expression network analysis identified co-expression modules significantly correlated with proline (r = 0.81) and malondialdehyde (r = 0.86) and selected the key hub gene Gh_A08G154500 (WRKY22), which was expressed 3.2 times higher in Jin than in TM-1 at 15 days of drought stress. Functional validation suggested that WRKY22 may form a “osmotic regulation–membrane protection” co-regulatory network by activating Pro synthesis genes (P5CS) and genes involved in the jasmonic acid signaling pathway. This study reveals, for the first time, the possible dual regulatory mechanism of WRKY22 in cotton’s drought resistance, providing a theoretical basis for cotton drought-resistant breeding.

## 1. Introduction

Cotton (*Gossypium* spp.) is not only one of the most important natural fiber crops in the world, but also an ideal material for studying genome evolution, polyploidization, and cell elongation [1]. A total of 35% of the world’s fibers come from cotton [2]. There are approximately 50 species in the genus Gossypium [3], and the long-domesticated tetraploid cotton, *Gossypium hirsutum*, has a large yield potential and wide adaptability, with over 95% of the cotton cultivation area planted with this species [4]. With global climate warming, freshwater scarcity, extreme weather such as high temperatures, and frequent droughts, drought has become a significant factor limiting cotton production [5]. Drought stress affects cotton yield and quality by altering metabolic activities and biological functions [6]. Ul-Allah et al. [7] reported that drought stress on cotton fiber development can lead to a yield loss of approximately 45%, and Abdelraheem et al. [8] found that water deficit during flowering reduces fiber strength, increases short fiber content, and lowers quality. Therefore, identifying drought-resistant genes in cotton, elucidating the molecular mechanisms of drought tolerance, and breeding drought-resistant cotton varieties are of significant importance for the textile industry.

Plants respond to drought stress through morphological, physiological, and biochemical changes [9]. They primarily resist drought stress by osmotic regulation, reactive oxygen species (ROS) scavenging, and maintaining cell membrane stability [10]. During this process, changes in the levels of certain substances are closely related to drought tolerance and can serve as important indicators for evaluating drought resistance in plants. When plants are subjected to drought stress, lipid peroxidation occurs, and malondialdehyde (MDA), the final product of lipid peroxidation [11], reflects the extent of damage to the plant cell membrane caused by environmental stress. Therefore, drought-resistant cotton varieties accumulate lower levels of MDA under drought conditions compared to sensitive varieties [12]. Proline (Pro) is a key osmotic regulator in plants [13]. When plants face drought stress, they produce large amounts of Pro by upregulating the expression of the synthesis enzyme gene *P5CS* [14], maintaining cell osmotic potential to resist drought stress. Superoxide dismutase (SOD) can scavenge ROS generated during drought stress [15], reducing lipid peroxidation levels. Chlorophyll (Chl) is an essential substance for photosynthesis in plants. Under drought stress, damage to the thylakoid structure of chloroplasts leads to a decrease in Chl, weakening photosynthesis [16]. Therefore, we selected MDA, Pro, SOD, and Chl as indicators to reflect the extent of drought stress in cotton.

Despite the identification of drought-responsive genes in cotton, such as *GhGABA-T* and *GhAO* mediating osmotic adjustment [17], and *GhMYB102* regulating ABA-dependent signaling [18], the systemic coordination of gene networks underlying drought adaptation remains elusive. Current transcriptomic studies predominantly focus on differential gene expression screening, yet fail to delineate hierarchical regulatory relationships or identify central hubs governing drought resistance traits. This fragmented understanding limits the development of precision breeding strategies targeting multigene synergies. For instance, while individual genes may enhance specific stress responses (e.g., ROS scavenging or stomatal regulation), their functional integration through co-expression modules, which is critical for balancing tradeoffs between stress tolerance and growth, is poorly characterized. Here, we employ a network-driven approach to dissect modular gene clusters strongly associated with drought-resistant phenotypes, aiming to uncover master regulators and their interactive mechanisms in cotton drought adaptation.

Currently, there is limited research on cotton drought resistance genes through physiological indicators under drought stress, RNA sequencing (RNA-Seq), and WGCNA (weighted gene co-expression network analysis). This study used multi-time point drought stress treatments on TM-1 (drought-sensitive) and Jin (drought-resistant) at the flowering and boll stage, dynamically monitoring the changes in physiological indicators such as MDA, Pro, SOD, and Chl in cotton leaves in response to drought stress. RNA-seq was used to construct gene expression profiles at different stress stages, WGCNA was employed to identify gene modules significantly associated with key physiological indicators of drought resistance, and Cytoscape was used for network visualization [19]. Finally, we identified core regulatory genes responsive to cotton drought stress. This study lays the foundation for further understanding of the molecular mechanisms of cotton drought resistance and developing new drought-resistant cotton varieties.

## 2. Experimental Methods

### 2.1. Planting and Treatment

TM-1 (sensitive to drought) and Jin (drought-resistant) were planted in 2024 in Huoyanghe City, Xinjiang Uygur Autonomous Region (44°20′ N–47°04′ N, 83°51′ E–85°51′ E). The experimental field was arranged with one film covering three rows (1 film 3 rows) and one film for planting (1 film 1 planting), each row 2 m long. Before the flower boll stage, they were managed according to normal irrigation and normal precipitation. In the boll stage of upland cotton, watering of the drought treatment area was stopped until harvest. Samples were taken from the third leaf from the top of the plant at 0, 7, 10, 15, and 25 days after drought stress, wrapped in foil, and stored in cryotubes. Immediately after collection, the samples were placed in liquid nitrogen and stored. Each group was repeated three times, and the samples were sent to NovoZhiyuan Ltd. (Tianjin, China). in Tianjin for library construction and sequencing.

### 2.2. Physiological Indicator Detection

Leaf samples were collected from two cotton cultivars, TM-1 and Jin. For analysis, three biological replicates were processed. Chl content was quantified using the Plant Chl Content Detection Kit (AKPL003M, Beijing Box Biotech, Beijing, China). Fresh leaf samples (0.1 g) were homogenized in 1 mL distilled water and 10 mg reagent, extracted with acetone in darkness for 3 h, and centrifuged. Absorbances at 663 nm (A663) and 645 nm (A645) were measured using a spectrophotometer/microplate reader (0.5 cm). The formulas for chlorophyll content (mg/g FW) were as follows:Chl a = [0.01 × (25.43A663 − 5.17A645) × D]/MChl b = [0.01 × (45.76A645 − 9.34A663) × D]/MTotal Chl = [0.01 × (40.59A645 + 16.09A663) × D]/M

(D: dilution factor; M: sample mass; V: 10 mL extract).

Pro content was determined using the Pro Detection Kit (AKAM003M): 50 mg tissue was homogenized in 1 mL 3% sulfosalicylic acid, boiled (95 °C, 10 min), reacted with acid ninhydrin reagent (100 °C, 30 min), and absorbance at 520 nm was measured after toluene extraction. MDA levels were assayed with the MDA Detection Kit (AKFA013M): 200 mg leaves were homogenized in 0.1% trichloroacetic acid, centrifuged (12,000× *g*, 15 min), reacted with thiobarbituric acid (95 °C, 30 min), and absorbance at 532, 600, and 450 nm was recorded for correction. SOD activity was measured using the SOD Activity Detection Kit (AKAO001M-50S): 100 mg tissue was homogenized in PBS (pH 7.8), centrifuged (15,000× *g*, 20 min), and 20 μL supernatant was mixed with WST-8 reagent; absorbance at 560 nm was kinetically monitored (10 min).

### 2.3. Library Preparation, Quality Control, and Sequencing

(1) Library preparation: polyadenylated mRNA was enriched using Oligo(dT) (Thermo Fisher Scientific’s, Waltham, MA, USA) magnetic beads, fragmented, and reverse-transcribed into cDNA. Libraries were constructed by end repair, adapter ligation, and PCR amplification (370–420 bp fragments selected via AMPure XP beads, Beckman Coulter’s, Brea, CA, USA); (2) quality control: libraries were quantified (Qubit 2.0 Fluorometer, Thermo Fisher Scientific’s, Waltham, MA, USA), assessed for insert size (Agilent 2100 Bioanalyzer, Agilent Technologies, Santa Clara, CA, USA), and validated for effective concentration (>1.5 nM) by qRT-PCR; (3) sequencing: pooled libraries were sequenced on an Illumina NovaSeq X Plus platform, Illumina Inc., San Diego, CA, USA (150 bp paired-end reads) using sequencing-by-synthesis (SBS), with fluorescence signals converted to sequence reads via standard protocols.

### 2.4. Data Quality Control

The image data of sequencing fragments obtained from the high-throughput sequencer are converted into sequence data (reads) through CASAVA base calling, and the file is in fastq format, which mainly contains the sequence information of the sequencing fragments and their corresponding quality information. The raw data obtained by sequencing contain a small number of reads with sequencing adapters or low sequencing quality. To ensure the quality and reliability of data analysis, the raw data need to be filtered. This mainly involves removing reads with adapters, reads containing N (N indicates undetermined base information), and low-quality reads (reads where more than 50% of bases have a Qphred score ≤ 5). At the same time, the clean data are evaluated for Q20, Q30, and GC content. Subsequent analyses are all performed based on the clean data for high-quality analysis.

### 2.5. Sequence Alignment to the Reference Genome

Using the upland cotton TM-1 reference genome (https://www.cottongen.org/species/Gossypium_hirsutum/CRI-TM1, accessed on 19 March 2025) [20], the index of the reference genome is constructed through HISAT2 v2.0.5 [21], and the paired-end clean reads are compared with the reference genome using HISAT2 v2.0.5.

### 2.6. Gene Expression Quantification

Feature Counts (1.5.0-p3) [22] was used to calculate the number of reads mapped to each gene. Then, the FPKM of each gene was calculated based on gene length, and the number of reads mapped to the gene was computed. FPKM refers to the expected number of transcript sequence fragments per kilobase of transcript length per million base pairs of sequencing. It also considers the effects of sequencing depth and gene length on read counts, and is currently one of the most widely used methods for estimating gene expression levels.

### 2.7. Differential Expression Analysis

For samples with biological replicates, DESeq2 software (1.20.0) [23] was used to perform differential expression analysis between two comparison groups. DESeq2 provides statistical programs for determining differential expression in count-based gene expression data using a model based on the negative binomial distribution. Genes with an adjusted *p*-value ≤ 0.05 discovered through DESeq2 were classified as differentially expressed. Before conducting differential gene expression analysis, the read counts for each sequencing library were adjusted using a normalization factor via the edgeR package (3.22.5) [24]. Differential expression analysis between two conditions was performed using the edgeR package (3.22.5). The *p*-value was adjusted using the Benjamini and Hochberg method [25], with the corrected *p*-value and |log2 fold change| used as thresholds for significant differential expression.

### 2.8. Differential Gene Enrichment Analysis

Differentially expressed gene GO enrichment analysis was performed using the clusterProfiler (3.8.1) software [26], with gene length bias corrected. GO terms with a corrected *p*-value less than 0.05 were considered significantly enriched by differentially expressed genes. KEGG is a database resource used to understand the high-level functions and utilities of biological systems, particularly from molecular-level information such as large-scale molecular datasets generated by genome sequencing and other high-throughput databases, including systems such as cells, organisms, and ecosystems. We used clusterProfiler (3.8.1) software to analyze the statistical enrichment of differentially expressed genes in KEGG pathways.

### 2.9. WGCNA

WGCNA is a systems biology approach used to describe gene association patterns between different samples. It can identify highly co-expressed gene sets, and, based on the internal connectivity of the gene set and its association with phenotypes, identify candidate biomarker genes or therapeutic targets. The R package WGCNA is a functional collection used for performing various weighted association analyses. It can be used for network construction, gene selection, gene cluster identification, topological feature calculation, data simulation, and visualization. Cytoscape (Cytoscape_v3.9.1) was used to visualize the gene modules.

### 2.10. qRT-PCR Validation

The drought-tolerant material Jin and drought-sensitive material TM-1 were selected as the extreme materials for candidate gene qRT-PCR analysis. Leaf samples were collected from field-grown cotton under natural drought conditions at 0 d, 7 d, 10 d, 15 d, and 25 d during the flowering and bolling stages. Total RNA was extracted using the RNAprep Pure Polysaccharide Polyphenol Plant Total RNA Extraction Kit (Tiangen, Beijing, China), and cDNA was synthesized using a reverse transcription kit (Tiangen). Cotton *Ubiquitin7* (*UBQ7*) was chosen as the reference gene [27]. Based on the reaction system provided by the Abm BlasTaq™ 2X qPCR Mix (dye) manual (Applied Biological Materials, Richmond, BC, Canada), the running program was set, and the relevant genes were amplified using the Applied Biosystems™ 7500 Fast Real-Time PCR Systems, Waltham, MA, USA (3 replicates). Relative expression levels were analyzed using the 2^−ΔΔCt^ method, and the specific primers are listed in Appendix A. Finally, GrandPad Prism version 10 [28] was used for data visualization.

## 3. Results

### 3.1. Identification of Drought Resistance of Jin and TM-1

To further compare the drought resistance levels between TM-1 and Jin, the physiological parameters of TM-1 and Jin under different drought stress conditions were measured (Figure 1). After drought stress, the MDA content in TM-1 leaves significantly increased with the intensification of drought stress, while the increase in MDA content in Jin leaves was relatively smaller. Chl and SOD content showed significant differences between TM-1 and Jin after 15 days of drought stress. Pro significantly increased in the leaves of both TM-1 and Jin under drought stress, and the Pro content in Jin leaves was significantly higher than in TM-1 with increasing drought stress. These results indicate that there are significant differences in drought resistance between Jin and TM-1. To further investigate the molecular mechanisms and candidate genes responsible for these differences in drought resistance, RNA-seq analysis was performed on leaf samples from Jin and TM-1 at five time points (0 d, 7 d, 10 d, 15 d, and 25 d) under drought stress.

### 3.2. RNA-Seq Data Analysis

In this study, 204.43 Gb of clean data were generated from 30 cotton leaf samples of TM-1 and Jin under drought stress at the flowering and boll-setting stages. Each sample had an average of 6.81 Gb of clean data, with Q30 base percentages exceeding 97.10%. Furthermore, the alignment rate of the reads to the reference genome ranged from 90.23% to 96.00%, with an average alignment rate of 94.60%. The quality metrics of the sequencing data, including raw read counts, Phred scores (Q30), and retention rates after filtering, are summarized in Appendix A. The Pearson correlation coefficient between the three biological replicates of the same sample showed correlations over 0.90 (Figure 2A). PCA results indicated that biological replicates are clustered together, and samples without drought stress clustered together, while samples under moderate drought stress (7 d and 10 d) clustered together, and samples under severe drought stress (15 d and 25 d) clustered together, suggesting that the transcriptomic data were highly reliable and reproducible (Figure 2B).

### 3.3. DEGs Analysis

To uncover the molecular mechanisms behind the drought resistance differences between different cotton varieties, we performed RNA-seq analysis on drought-stressed samples and detected a total of 81,679 expressed genes. DEGs under drought stress were then identified using stringent thresholds (FDR < 0.05, |log2FC| ≥ 1). As shown in Figure 3A, the sensitive variety TM-1 exhibited 13,439 DEGs (6687 upregulated/6752 downregulated) during the early stress stage (0–7 d), while the drought-resistant variety Jin maintained a high response level during prolonged stress (15–25 d) with 6204 DEGs (2409 upregulated/3795 downregulated). Notably, Jin showed a significantly higher number of DEGs (2927) during the mid-stress stage (7–10 d) compared to TM-1 (469, *p* < 0.01), suggesting that Jin may possess the characteristic of continuously activating drought resistance pathways. Venn analysis revealed that out of 25,123 DEGs, 12,406 were common genes, and the number of unique genes in Jin (8544) was 2.05 times that of TM-1 (4172). This difference may explain the divergence in drought resistance between the varieties: Jin maintains cell homeostasis by continuously regulating the synthesis of osmotic protectants, while the TM-1 response mechanism is limited to the early stage of stress.

### 3.4. KEGG and GO Enrichment Analysis of Differentially Expressed Genes Shared by Jin and TM-1

To explore the coordinated response mechanisms of the drought-resistant material Jin and the drought-sensitive material TM-1 under drought stress, we performed a systematic analysis of the 12,406 common DEGs. Through temporal expression pattern clustering, DEGs were divided into four characteristic modules (Figure 4A): Cluster 1 genes were lowly expressed under no stress and significantly upregulated as stress duration increased (15–25 d); Cluster 2 and Cluster 4 genes were highly expressed under no stress, but exhibited TM-1-specific (Cluster 2) and Jin-specific (Cluster 4) expression advantages; Cluster 3 genes were activated early in the stress period and maintained a stable expression level throughout the stress stages. KEGG enrichment analysis revealed that DEGs mainly participated in metabolic regulation (Figure 4B), including lipid metabolism (fatty acid elongation, glycerophospholipid metabolism), carbohydrate metabolism (fructose/mannose metabolism), amino acid metabolism (alanine/aspartate/glutamate metabolism), secondary metabolism (steroid/carotenoid biosynthesis), and transport processes (ABC transporter proteins). Notably, stress response pathways closely related to ascorbate metabolism, sulfur metabolism, and redox regulation were significantly enriched. GO analysis (Figure 4C) showed that biological processes primarily involved amino acid biosynthesis (cell/α-amino acids/sulfur-containing amino acids), polysaccharide metabolism (glucan synthesis), and catabolism (aromatic amino acids); cellular components enriched in structures related to genetic information regulation (chromatin/nucleosome) and stress response sites (cell wall/extracellular matrix); molecular functions were prominent in metabolic catalytic activities (isomerases/transferases/oxidoreductases) and molecular binding capabilities (unfolded proteins/ssDNA). These findings systematically reveal that plants may adapt to drought stress through a multidimensional molecular regulatory network.

### 3.5. Jin- and TM-1-Specific DEGs KEGG and GO Enrichment Analysis

Through functional analysis of drought-related DEGs in Jin and TM-1 cotton varieties, it was found that the 8544 DEGs specifically expressed in Jin (|log2FC| > 1, FDR < 0.05) were dynamically divided into four clusters based on drought stress duration (Figure 5A): Cluster 1 (high expression at 0 days control), Cluster 2 (high expression at 15 days drought), Cluster 3 (high expression at 25 days drought), and Cluster 4 (high expression at 7 and 10 days drought). The KEGG pathways of these DEGs were significantly enriched in lipid metabolism (ether lipids, alpha-linolenic acid metabolism), stress response signaling (MAPK, phosphoinositide signaling), and secondary metabolism (isoquinoline alkaloid, sesquiterpene biosynthesis), while GO functions were involved in microtubule binding, jasmonic acid biosynthesis, and the outer mitochondrial membrane (Figure 5B,C). In TM-1, among the 4172 specific DEGs, Cluster 2 (1870 genes) was highly expressed under control conditions, and suppressed under drought stress. The metabolic characteristics of this cluster were focused on energy metabolism (TCA cycle, glycolysis), DNA repair (non-homologous end joining), and ion channel activity (Figure 5D). The comparison between the two varieties indicated that Jin might maintain membrane stability through lipid metabolism and microtubule-related mechanisms, while TM-1 relies on energy supply and genomic stability to cope with drought, revealing the molecular differentiation of drought resistance strategies between them.

### 3.6. Transcription Factors Analysis

We submitted the genomic sequences of the 25,123 identified DEGs to the PlantTFDB database (http://planttfdb.cbi.pku.edu.cn/; accessed on 9 November 2024) for transcription factor prediction, and a total of 1551 transcription factors were identified. These transcription factors were classified into different families, including MYB, ERFs, bHLHs, C2H2s, NAC, and WRKYs (Figure 6A). To further understand the function of these 1551 transcription factors, we used the k-means clustering algorithm to cluster the 1551 differentially expressed transcription factors into 4 clusters, and then annotated the functional categories of each cluster using KEGG (Figure 6B).

Cluster 1 was highly expressed without drought stress and showed a decrease in expression under drought stress. It was mainly annotated in nitrogen metabolism, isoquinoline alkaloid biosynthesis, betaine biosynthesis, glycine, serine, threonine metabolism, and tyrosine metabolism. Cluster 2 was highly expressed under drought stress at 15 days and was mainly annotated in linolenic acid metabolism, the MAPK signaling pathway, N-glycan biosynthesis, galactose metabolism, and brassinosteroid biosynthesis. Cluster 3, in contrast to Cluster 1, was lowly expressed without drought stress, but showed an increase in expression under drought stress. It was mainly annotated in ribosome biogenesis in eukaryotes; RNA polymerase; RNA degradation; valine, leucine, and isoleucine degradation; and nucleotide excision repair. These results suggest that transcription factors coordinate responses to drought stress through the regulation of lipid metabolism, osmotic regulation (such as betaine), and signaling pathways (such as MAPK).

### 3.7. WGCNA

To study the gene regulatory network involved in cotton under drought stress, we constructed a co-expression network of 25,123 DEGs in the cotton lines using WGCNA (β soft-thresholding value of seven, scale-free R2 > 0.8) (Figure 7A). Thirteen co-expression modules were obtained, with different colors representing different modules. The correlation between the modules and physiological indicators was calculated, and two significantly highly correlated modules were identified: the blue module, which was significantly correlated with Pro (r = 0.81, *p* < 0.01), and the yellow–green module, which was significantly correlated with MDA (r = 0.86, *p* < 0.01) (Figure 7B). The hub genes within the blue and yellow–green modules were selected using Cytoscape, and the three genes with the highest connectivity were identified as hub genes. A total of six hub genes were obtained, and the top 100 connections within the blue and yellow–green modules were visualized using Cytoscape based on weight values (Figure 7C–F). Among the six hub genes, *Gh_A08G154500* encodes a WRKY transcription factor, *Gh_A05G348000* encodes a bromodomain-containing protein BET subfamily transcription factor, *Gh_A11G336600* encodes protein kinase C conserved region 2 (*CalB*), *Gh_A12G140900* encodes a cotton sugar and high-affinity sucrose synthase as well as a sucrose and Gol-specific galactoside hydrolase activity protein, *Gh_A11G122800* encodes a drought-related protein *PCC13-62*, and *Gh_A11G283900* encodes a protein similar to the antifungal chitin-binding protein hevein from rubber tree latex.

### 3.8. qRT-PCR

qRT-PCR was used to detect the expression patterns of these six candidate genes in the leaves of TM-1 and Jin under different drought stress conditions. The expression levels of *Gh_A11G122800* and *Gh_A11G283900* significantly increased after 7 days of drought stress, while the expression levels of *Gh_A08G154500*, *Gh_A05G348000*, *Gh_A11G336600*, and *Gh_A12G140900* significantly increased after 15 or 25 days of drought stress. Furthermore, there were significant differences in the expression levels of these six candidate genes at the same time point after drought stress between TM-1 and Jin, indicating that the expression patterns of these six candidate genes were different under drought stress (Figure 8A). The results also suggested that these genes participate in the regulation of cotton drought stress through different expression patterns. Moreover, the expression trends of these six candidate genes under drought stress were consistent with the transcriptome data, further confirming the reliability of the transcriptome sequencing results (Figure 8B).

## 4. Discussion

### 4.1. Physiological Response and Molecular Regulatory Network Differentiation Between Drought-Tolerant and Drought-Sensitive Materials

This study systematically compares the dynamic changes in physiological indicators and transcriptomic responses of the drought-tolerant variety Jin and the drought-sensitive variety TM-1 under drought stress, revealing significant differentiation in their drought resistance mechanisms. Physiological phenotype analysis shows that Jin efficiently regulates osmotic balance (continuous accumulation of Pro) and antioxidant defense (SOD activity maintenance) systems under drought stress, significantly reducing membrane lipid peroxidation levels (low MDA accumulation), while delaying chloroplast damage (small decrease in Chl), thus maintaining cell homeostasis [29]. In contrast, TM-1 exhibits significant membrane system collapse (MDA surge) and loss of photosynthetic capacity (sharp decline in Chl) in the later stages of stress, suggesting a lack of sustained and systematic drought resistance mechanisms [30]. This physiological phenotype difference is further supported at the transcriptomic level: Jin activates a large number of DEGs (2927) during the mid-stress stage (7–10 d), significantly more than TM-1 (469), and its unique DEGs (8544) are double that of TM-1 (4172). This difference in gene expression dynamics indicates that Jin possesses an earlier, broader molecular response network that continuously regulates lipid metabolism (α-linolenic acid metabolism, ether lipid synthesis) and secondary metabolism (sesquiterpene, isoquinoline alkaloid biosynthesis) pathways to enhance membrane stability and stress signal transmission efficiency. In contrast, TM-1’s molecular response is focused on early-stage energy metabolism (TCA cycle, glycolysis) and DNA repair mechanisms, which may result in physiological breakdown in the later stages due to a failure to activate osmotic protection systems in time [31].

The analysis of transcription factors further revealed the complexity differences in the regulatory networks between the two varieties. The WRKY, NAC, and MYB family members specifically upregulated in Jin during the mid-stress period (7–15 d) were significantly enriched in the MAPK signaling and jasmonic acid biosynthesis pathways. These transcription factor families have been widely reported to enhance drought resistance by regulating the synthesis of osmotic substances and the expression of antioxidant genes [32,33,34]. In contrast, the suppressed bHLH and C2H2 family members (cluster 2) in TM-1 are related to ion channel activity and DNA repair [35], with its drought resistance strategy focusing more on maintaining genomic stability rather than actively adapting to the stress environment. Notably, cluster 3 (stress-induced) transcription factors in Jin were significantly enriched in ribosome biosynthesis and RNA degradation pathways, suggesting that they may rapidly reshape the proteome by regulating translation efficiency to adapt to drought. This mechanism has also been reported in Arabidopsis and rice [36,37], but it is being revealed for the first time in cotton. In summary, the differences in drought resistance between Jin and TM-1 are not only reflected in the dynamic physiological responses, but also arise from significant differentiation in the breadth, timing, and functional integration efficiency of their molecular regulatory networks.

### 4.2. WGCNA of Co-Expressed Modules and Functional Validation of Hub Genes

The gene co-expression network constructed based on WGCNA for the first time dynamically correlates the gene modules related to cotton drought resistance with key physiological indicators, breaking through the limitations of traditional differential gene screening. The blue module (r = 0.81, *p* < 0.01) was highly positively correlated with Pro content, with the core hub gene *Gh_A12G140900* encoding cotton raffinose synthase, which catalyzes the synthesis of raffinose family oligosaccharides (RFOs), substances that not only serve as osmotic regulators, but also enhance cell dehydration tolerance by protecting membrane integrity [38]. Another hub gene, *Gh_A11G122800*, encodes a drought-related protein, *PCC13-62*, which stabilizes cell membrane structures to prevent dehydration damage during plant recovery [39]. Its homologous gene’s function in cotton has been reported for the first time. The green–yellow module (r = 0.86, *p* < 0.01) was significantly correlated with MDA content, and its hub gene *Gh_A05G348000* (BET bromodomain protein) might regulate the expression of ROS-related genes through epigenetic regulation [40]. While *Gh_A11G336600* (protein kinase C) might activate membrane lipid repair enzymes via phosphorylation modification [41]. These findings directly link traditional physiological indicators to molecular regulatory networks, offering a new perspective on the modular regulatory mechanisms underlying cotton drought resistance.

qRT-PCR validation showed that the expression patterns of the six candidate Hub genes were highly consistent with the RNA-seq data (Figure 8A,B), and there were significant expression differences between varieties. For example, *Gh_A08G154500* (*WRKY22*) showed a 3.2-fold higher expression in Jin compared to TM-1 at 25 days of drought (*p* < 0.001), with its peak expression coinciding with the trend of Pro accumulation, suggesting that it may enhance osmotic regulation by positively regulating the Pro synthesis pathway [42]. It is noteworthy that *Gh_A11G283900* (chitinase-binding protein homolog) was significantly upregulated (2.8-fold, *p* < 0.01) in Jin at 7 days of drought, while no significant change was observed in TM-1. This gene may help resist the mechanical damage caused by osmotic stress by enhancing cell wall rigidity [43]. This spatiotemporal expression specificity reflects the functional differentiation of Hub genes in drought resistance strategies between varieties, providing target combinations for multi-gene collaborative breeding.

### 4.3. WRKY22 Enhances Cotton Drought Resistance Through a Multidimensional Regulatory Network

This study found that the WRKY transcription factor *Gh_A08G154500* (*WRKY22*) was continuously expressed at high levels in Jin after drought, and its expression dynamics were significantly correlated with Pro accumulation and MDA inhibition, indicating its core role in cotton drought resistance regulation. Previous studies have shown that WRKY family members can directly activate key Pro synthesis genes, such as *P5CS* (1-pyrroline-5-carboxylate synthetase), by binding to the W-box cis-element [44]. In this study, the expression of *P5CS* in Jin was 1.8 times higher than that in TM-1 at the mid-stress period (10 d), consistent with the expression trend of *WRKY22*. Further analysis revealed that the co-expression module of *WRKY22* (blue module) was enriched in MAPK signaling pathways and jasmonic acid biosynthesis genes. It may amplify stress signals through the MAPK-WRKY cascade [45], while also cooperating with the jasmonic acid pathway to regulate the antioxidant system (SOD activity) to reduce MDA production [46]. This regulatory pattern is similar to the mechanism by which Arabidopsis *AtWRKY57* enhances drought resistance through JA signaling [47].

Additionally, *WRKY22* may coordinate physiological responses through a dual mechanism: on one hand, it directly activates the Pro synthesis pathway to maintain cellular osmotic potential [48]; on the other hand, it enhances membrane lipid repair ability by regulating downstream genes such as *Gh_A11G336600* (protein kinase C). This “osmotic membrane protection” dual-module regulatory network may be key to Jin’s stronger drought resilience compared to TM-1. It is worth noting that the expression of *WRKY22* in TM-1 decreased (0.6-fold, *p* < 0.05) at 25 days of drought, which is consistent with the phenotype of a dramatic increase in MDA content, indicating a functional defect in the regulatory network of this gene in the drought-sensitive variety. Future research can systematically analyze the regulatory effects of *WRKY22* on Pro metabolism, ROS clearance, and membrane stability by CRISPR knockout/overexpression, and use molecular biology techniques to identify upstream and downstream regulatory genes, in order to improve the *WRKY22* regulatory network model for cotton drought resistance (Figure 9).

## 5. Conclusions

This study deciphers drought resilience mechanisms in cotton by integrating physiological and transcriptomic dynamics between a tolerant (Jin) and a sensitive (TM-1) cultivar. Jin’s superior drought adaptation stems from sustained osmotic regulation (Pro accumulation) and redox balance (elevated SOD), mitigating oxidative damage. Transcriptomics revealed Jin’s unique activation of lipid metabolism and stress signaling to stabilize membranes and coordinate responses. A co-expression network identified a WRKY transcription factor *Gh_A08G154500* (*WRKY22*) as a central hub, linking Pro biosynthesis and jasmonate signaling to form an “osmotic membrane protection” axis. These insights advance the molecular breeding of drought-tolerant cotton by prioritizing actionable targets like WRKY regulators for crop improvement.

## Figures and Tables

**Figure 1 plants-14-01407-f001:**
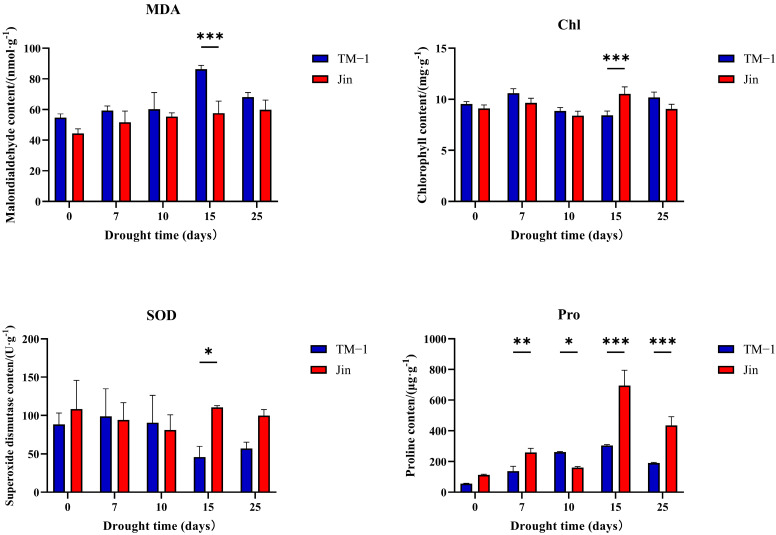
Physiological biomarker dynamics under progressive drought stress. Statistical analysis of malondialdehyde (MDA), chlorophyll (Chl), superoxide dismutase (SOD), and proline (Pro) content in drought-sensitive TM-1 and drought-tolerant Jin under different drought stress conditions. Stress phases: 0 d (pre-stress), 7 d, 10 d, 15 d, and 25 d. Significance determined by one-way ANOVA (* *p* < 0.05, ** *p* < 0.01, *** *p* < 0.001; n = three biological replicates).

**Figure 2 plants-14-01407-f002:**
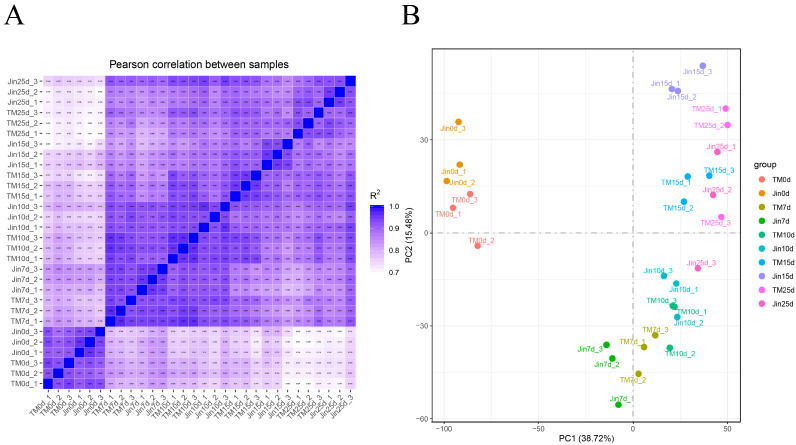
Transcriptomic divergence between cultivars revealed by correlation and principal component analyses. (**A**) Inter-cultivar transcriptome coordination: pairwise Pearson correlation coefficients (r) of RNA sequencing (RNA-seq) profiles between drought-sensitive TM-1 and drought-tolerant Jin under progressive drought. Stress phases: (0 d, 7 d, 10 d, 15 d, and 25 d). (**B**) Dimensionality reduction mapping: principal component analysis (PCA) of transcriptomic profiles with variance decomposition (PC1: 38.72%, PC2: 15.48%).

**Figure 3 plants-14-01407-f003:**
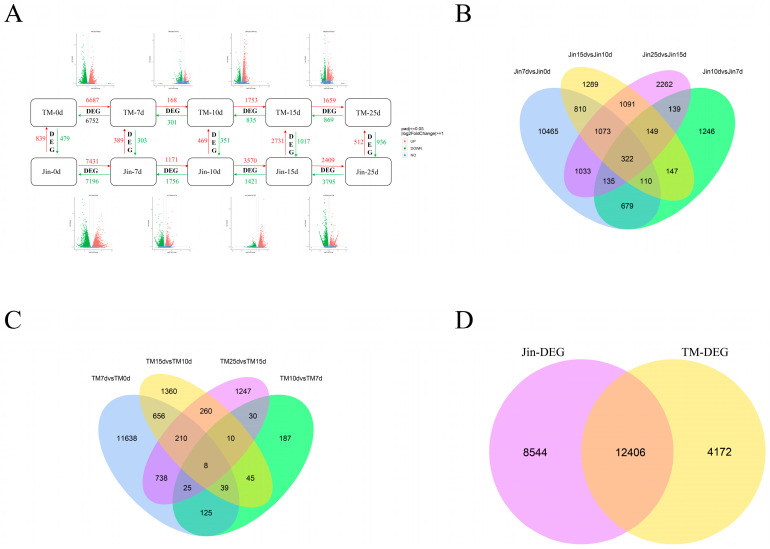
Stage- and cultivar-specific divergence of drought-responsive gene expression. Expressed genes (DEGs) defined by |log2(fold change)| > 1, FDR < 0.05. (**A**) Number of upregulated and downregulated DEGs in Jin and TM-1 under drought stress. (**B**) Number of DEGs in Jin at different stages of drought stress. (**C**) Number of DEGs in TM-1 at different stages of drought stress. (**D**) Number of DEGs between Jin and TM-1.

**Figure 4 plants-14-01407-f004:**
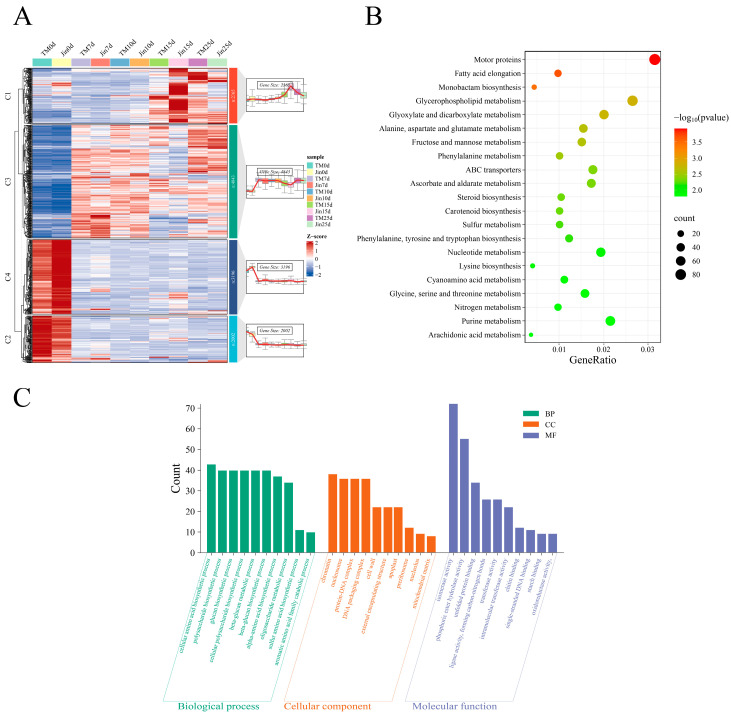
Shared drought-responsive genes: expression clusters and functional annotations. (**A**) Expression patterns of shared DEGs identified by hierarchical clustering (left), with the top five most significantly enriched Kyoto Encyclopedia of Genes and Genomes (KEGG) pathways (lowest *p*-values) displayed for each cluster (right). (**B**) KEGG pathway enrichment analysis of shared DEGs. (**C**) Gene ontology (GO) enrichment analysis of shared DEGs, showing the top 10 terms for biological processes (BP), molecular functions (MF), and cellular components (CC).

**Figure 5 plants-14-01407-f005:**
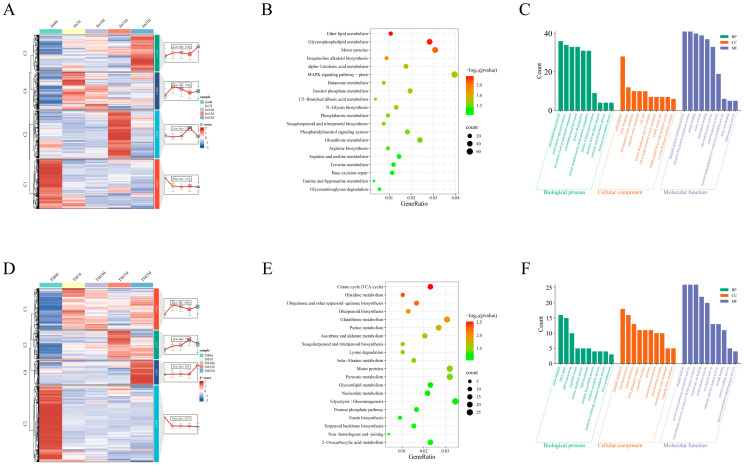
Expression and functional annotation of cultivar-specific drought-responsive genes. (**A**) Expression patterns of Jin-specific DEGs identified by hierarchical clustering (left), with the top five most enriched KEGG pathways (lowest *p*-values) annotated for each cluster (right). (**B**) KEGG pathway enrichment of Jin-specific DEGs. (**C**) GO enrichment of Jin-specific DEGs, showing the top 10 terms for BP, MF, and CC. (**D**) Expression patterns and functional annotation of TM-1-specific DEGs. Hierarchical clustering results (left) and top five KEGG pathways per cluster (right) are displayed. (**E**) KEGG pathway enrichment of TM-1-specific DEGs. (**F**) GO enrichment of TM-1-specific DEGs, highlighting the top 10 terms for BP, MF, and CC.

**Figure 6 plants-14-01407-f006:**
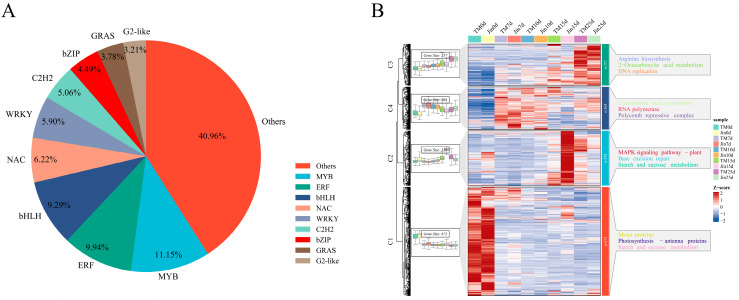
Composition and functional analysis of differentially expressed transcription factor families. (**A**) Proportion of differentially expressed transcription factor families. (**B**) Expression patterns and functional enrichment analysis of differentially expressed transcription factors. The KEGG annotation results for each cluster are shown on the right, displaying the top five pathways with the smallest *p*-values.

**Figure 7 plants-14-01407-f007:**
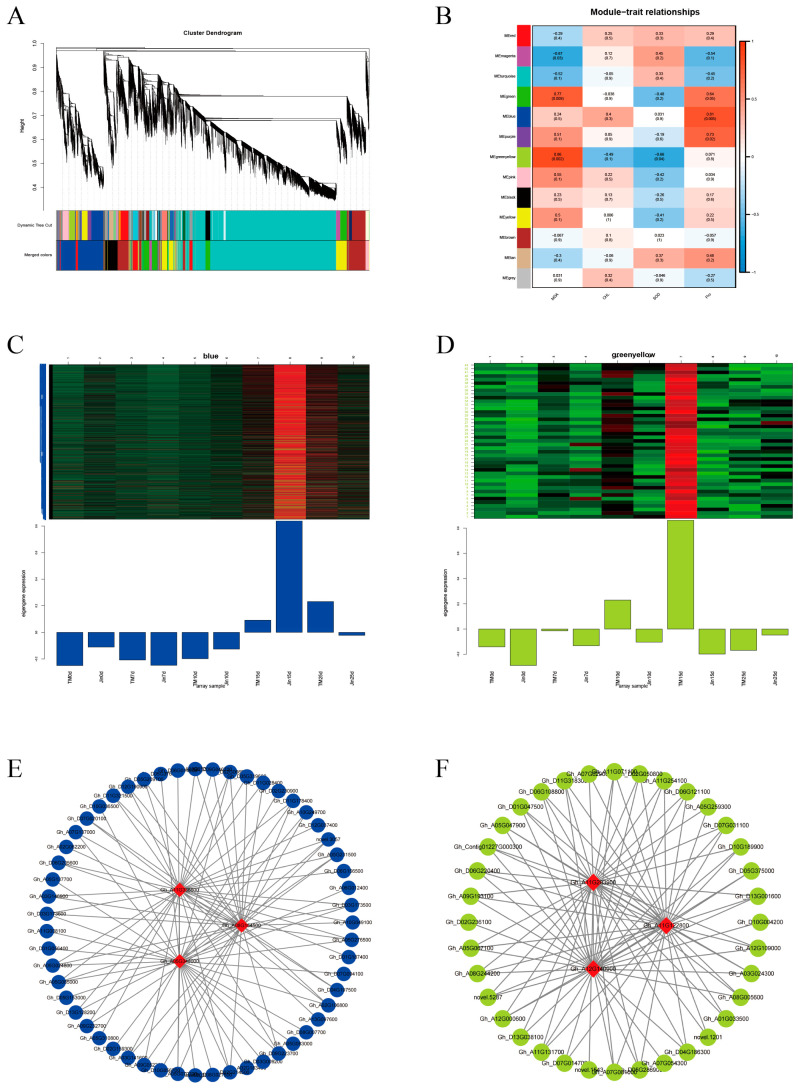
Co-expression network modules and hub regulatory genes under drought stress. (**A**) Hierarchical clustering dendrogram of weighted gene co-expression network analysis (WGCNA) modules; colors represent distinct co-expression modules (β soft-thresholding power = seven, scale-free topology model fit R^2^ > 0.8). (**B**) Module–trait correlation heatmap: correlations between modules and physiological indicators. Red indicates positive correlation, blue indicates negative correlation. (**C**) Expression trends of genes in the blue module across drought stages (0 d, 7 d, 10 d, 15 d, 25 d). (**D**) Expression trends of genes in the yellow–green module across drought stages. (**E**,**F**) Hub gene interaction networks: top 100 weighted connections for hub genes (node size reflects connectivity).

**Figure 8 plants-14-01407-f008:**
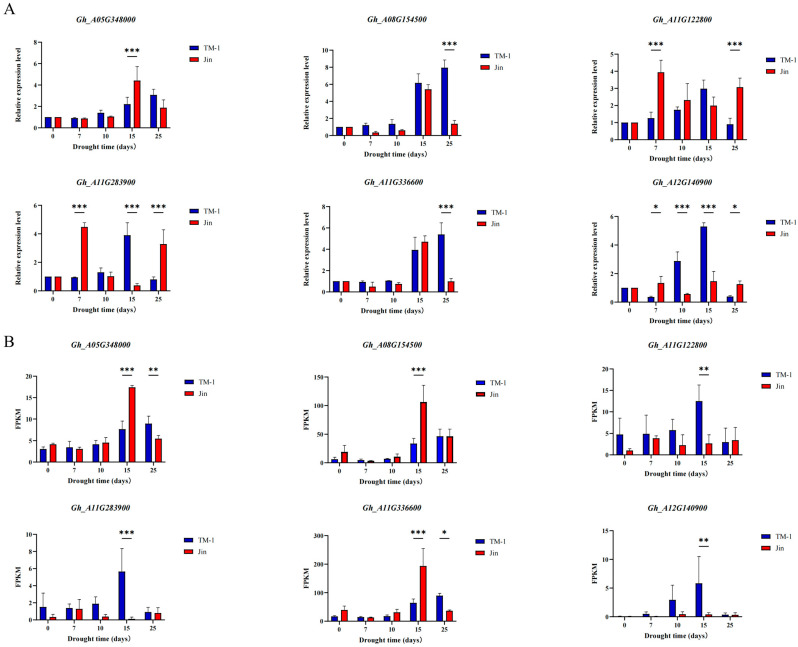
Validation of candidate gene expression under progressive drought stress. (**A**) qRT-PCR analysis: expression patterns of candidate genes at different drought durations (0 d, 7 d, 10 d, 15 d, and 25 d) analyzed by quantitative reverse transcription PCR (qRT-PCR). Error bars indicate standard error (SE) of three biological replicates. Significance levels: * *p* < 0.05, ** *p* < 0.01, *** *p* < 0.001 (Student’s *t*-test). (**B**) RNA-seq validation: transcript abundance of candidate genes quantified by fragments per kilobase million (FPKM) values.

**Figure 9 plants-14-01407-f009:**
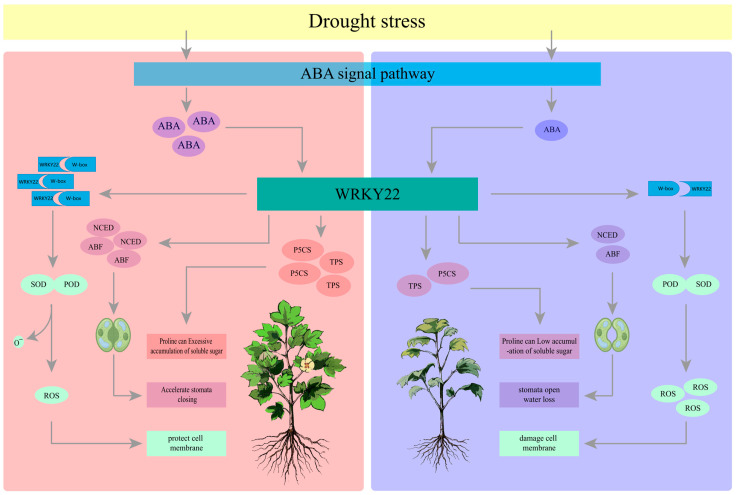
*WRKY22*-mediated drought regulatory network integrates physiological and molecular responses.

## Data Availability

The original RNA-seq data have been uploaded to the NCBI_SRA database (PRJNA1240467).

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
