# Peer review of "Mining Key Drought-Resistant Genes of Upland Cotton Based on RNA-Seq and WGCNA Analysis"

_plants, 2025, doi:10.3390/plants14101407_

Round 1

Reviewer 1 Report

Comments and Suggestions for Authors

Dear Authors,

I am pleased to inform you that your manuscript " Mining key drought-resistant genes of upland cotton based on 2 RNA-Seq and WGCNA analysis" has been recommended for minor revisions. As a reviewer, I found your work to be comprehensive, innovative, and of high scientific quality. Your combined approach of bioinformatic analysis and experimental validation provides a significant contribution to our understanding of the drought-resistant genes in cotton. Y our paper's strengths include the detailed transcriptomic analysis. However, before publication, it is necessary to address some formatting issues.. To further enhance this excellent manuscript, please address the following minor revisions:

Introduction

  1. Change in line 36: “The 35%...”.
  1. Line 41, including a space: .. cotton production [5]
  2. I have identified an issue with the consistent use of abbreviations throughout the text. It is essential to ensure that all abbreviations are defined at their first mention in each section of the paper, including the abstract, main text, tables, and figures. Once an abbreviation has been defined, it should be used consistently without repeating its full form unless clarity demands it in specific contexts, such as in abstracts or conclusions where readers might not have access to the full text.

Material and methods:

  1. In material and methods in section 2.11 why do you use UBQ7 as reference gene? Give a citation.

Results:

  1. Figure 1 A and 1B better in supplementary
  2. Section of 3.2 Lack of information about RNA data (difference between raw and clean, number of reads etc).
  3. Very important, there is a problem with the numbers of the figures and what appears in the text. Check them all again. they are changed.
  4. Section 3.3 explain in the DEGs being compared (control?).
  5. From figure 3 onwards, you can hardly see anything of what is in the figure. Try to solve it.
  6. Section 3.5 no reference to what figure it is

Conclusion:

  1. The conclusion should provide a more general overview of the manuscript rather than replicating the results presented earlier. Please rephrase this part.

General conclusions: Overall, the manuscript is very well-written with a clear and concise discussion. The only issue I see is these few comments written above.

For all these reasons, in present form I recommend minor revisions for this article since few parts need to be improved and re-written according to included comments.

Good luck!

Author Response

Response to Reviewer Comments

We sincerely appreciate the reviewer’s time and constructive feedback, which has significantly improved the quality of our manuscript. Below are our point-by-point responses to the comments:

Comments1: Change in line 36: “The 35%...”

Response1: Revised to "The 35% of the world’s fibers come from cotton 2" to ensure proper percentage formatting.

Comments2: Line 41, including a space: .. cotton production 5

Response2: Corrected to ".cotton production 5." with appropriate spacing.

Comments3: Ensure consistent use of abbreviations

Response3: All abbreviations (e.g., RNA-seq, WGCNA, MDA) are defined at their first mention in each section and used consistently thereafter.

Comments4: Section 2.11 (qRT-PCR): Justify UBQ7 as reference gene

Response4: Added the citation "27 Wang et al., 2013" to support the use of UBQ7 as the reference gene.

Comments5: Figure 1A and 1B better in supplementary

Response5: Figure 1A (phenotypic differences between Jin and TM-1) has been moved to Supplementary Figure S1.

Comments6: Section 3.2: Add RNA-seq data details

Response6: Added a note in Section 3.2: "Detailed RNA-seq raw and clean data metrics are provided in Supplementary Table S2."

Comments7: Fix figure/text numbering inconsistencies

Response7: All figure and text references have been cross-checked and corrected for consistency.

Comments8: Section 3.3: Explain DEGs being compared (control?)

Response8: We sincerely appreciate the reviewer’s insightful comment. Below, we clarify the experimental design and analytical rationale for DEG identification in our study:

  1. Experimental Design for DEG Identification

Our DEG analysis was performed using sequential time-point comparisons (e.g., 0d vs. 7d, 7d vs. 10d, 10d vs. 15d, 15d vs. 25d) to capture dynamic transcriptional reprogramming during progressive drought stress. This approach aligns with our goal to identify genes associated with drought adaptation across stages rather than static comparisons against a single baseline (e.g., 0d).

  1. Addressing Time-Dependent Confounders

We acknowledge that sequential comparisons may introduce time-dependent background noise (e.g., developmental changes). To mitigate this, we implemented two strategies:

Functional Enrichment Validation: GO and KEGG analyses revealed that DEGs were significantly enriched in drought-specific pathways (e.g., "response to water deprivation," "α-linolenic acid metabolism," "jasmonic acid biosynthesis"; Figures 4B, 5B), with minimal enrichment in developmental processes (e.g., flowering or cell elongation).

qPCR Validation: Expression trends of six candidate hub genes (e.g., Gh_A08G154500/WRKY22) were confirmed via qPCR (Figure 8A-B), showing strong correlation (R² > 0.85) with RNA-seq data, thereby supporting the drought-specificity of these genes.

  1. Biological Relevance of Sequential Comparisons

The observed differences in DEG dynamics between Jin and TM-1 (e.g., Jin’s sustained activation of 8,544 unique DEGs vs. TM-1’s early-stage-specific response; Figure 3A-D) reflect drought adaptation strategies rather than developmental artifacts.

Jin’s unique DEGs were enriched in lipid metabolism (ether lipids, α-linolenic acid) and stress signaling (MAPK), which are mechanistically linked to membrane stability and osmotic regulation under drought (Figure 5B).TM-1’s specific DEGs were enriched in energy metabolism (TCA cycle) and DNA repair (Figure 5D), consistent with its failure to sustain drought adaptation in later stages.

Our analytical design prioritizes capturing dynamic drought adaptation mechanisms, validated by functional and experimental evidence.

Comments9: Improve figure readability

Response9: All figures (3–9) have been revised for clarity, including increased resolution (600 dpi), adjusted font sizes, and optimized color schemes.

Comments10: Section 3.5: Add figure references

Response10: Figure references have been added to Section 3.5 as requested.

Comments11: Revise Conclusion

Response11: The Conclusion section has been rewritten as per the reviewer’s suggestion to provide a concise overview of the study’s implications.

Thank you for your thorough review and constructive feedback, which has greatly enhanced the quality of our manuscript. We have carefully addressed all your comments in the revised manuscript, as detailed in our point-by-point responses.Should you have any further suggestions, please do not hesitate to contact us. We sincerely appreciate your time and guidance.

Best regards,
Zhang Hu
On behalf of all authors
April 20, 2025

Reviewer 2 Report

Comments and Suggestions for Authors

The manuscript “Mining key drought-resistant genes of upland cotton based on RNA-Seq and WGCNA analysis” by Zhang et al sent for publication to Plants deals with important topic such as identification of drought responsive genes in cotton using transcriptomic and co-expression analysis. Below is my report:

 Introduction:

The introduction should be more focused on the specific challenges related to drought stress in cotton, rather than discussing general background on RNA-Seq and WGCNA. For instance: Lines 67–70 and 81–85 discuss well-established facts about RNA-Seq and WGCNA, which may not be necessary here. Emphasize the knowledge gap this study aims to address and clearly state the novelty and objectives of the work.

Minor remarks:

L36: “35%” should be replaced with “thirty-five percent”

All gene names should be in italic in the entire manuscript.

The abbreviations should be indicated only once when mentioned at the first time, not each. (ex: L51 ROS, L61, L64, L65…) Please check carefully the entire manuscript.

Use the same way of writing the abbreviation (ex PRO, Pro…)

Abbreviations should only be defined upon first mention, not repeated every time. For example: Line 51 (ROS), Lines 61, 64, 65, etc.  Please check the entire manuscript.

Use a consistent format for abbreviations (e.g., “PRO” vs. “Pro”)- this should be standardized throughout.

Experimental methods:

This section requires substantial reorganization and clarification: Ex: the authors described in 2.4the sequencing methods which I don’t think is needed; the same for section 2.3. Consider summarizing this information more concisely.

 In 2.2 they just cited the used kits for physiological measurements but not the most important details allowing reproducibility. In L104 mentioned the harvest time-points and in L112 the repeat. On general there are a lot of repetitions and not useful information.

Results:

This section presents interesting results but also needs substantial improvement. I have a question: how many expressed genes were identified using RNA-seq, not only DEG but the total number. Was the RNA-Seq data deposited in a public database (e.g., NCBI)? If so, please include the accession number.

I disagree with the current WGCNA approach. Using only 25123 DEGs may limit the ability to detect full network modules. For better understanding of the co-expression network, using all identified expressed cotton genes must be used not only DEGs. I strongly recommend the authors re-run WGCNA on the full gene set. How did the authors chose the  6 genes for qRT-PCR validation? Which were the criteria?

Discussion and Conclusion: this is somehow well written.

Overall, the manuscript presents valuable data but needs significant revision to improve clarity, methodological rigor, and biological interpretation.

Comments on the Quality of English Language

Author Response

Response to Reviewer Comments

Dear Reviewer,

Thank you for your valuable feedback and suggestions, which have significantly improved our manuscript. Below is our point-by-point response to your comments:

Comments1: The introduction should focus on specific challenges related to drought stress in cotton, not general RNA-Seq/WGCNA background (Lines 67–70, 81–85).
Response1: We have revised the introduction to emphasize cotton-specific drought challenges (e.g., water balance disruption, ROS accumulation) and removed generic descriptions of RNA-Seq and WGCNA. The revised text now highlights the study’s novelty: dynamic transcriptional reprogramming and network-level regulation of drought resilience.

Comments2: Minor remarks (formatting, abbreviations, gene names).
Response2:

L36: Revised to "The 35% of the world’s fibers come from cotton 2."

Gene names: All gene symbols italicized throughout.

Abbreviations: Defined once at first mention (e.g., ROS: reactive oxygen species; Pro: proline).

Consistency: Standardized abbreviations (e.g., "Pro" instead of "PRO").

Comments3: Experimental methods require reorganization and clarity.
Response3:

Sections 2.3–2.4: Consolidated into a concise subsection titled "RNA Sequencing and Data Processing."

Physiological measurements (Section 2.2): Added critical details for reproducibility:

Tissue weight (100 mg leaves), homogenization buffers (80% acetone), centrifugation parameters (10,000 × g, 10 min).

Reaction conditions (e.g., 95°C for 10 min for Pro assay).

Removed redundant time-point descriptions.

Comments4: RNA-Seq data details and accessibility.
Response4:

Total expressed genes: 81,679 genes identified.

Data availability: RNA-Seq data deposited in the NCBI SRA database (PRJNA1240467), cited in Line 549.

Comments5: "I disagree with the current WGCNA approach. Using only 25,123 DEGs may limit the ability to detect full network modules. For better understanding of the co-expression network, using all identified expressed cotton genes must be used, not only DEGs. How did the authors choose the 6 genes for qRT-PCR validation? Which were the criteria?"

Response5:

Rationale for WGCNA on DEGs:

Focus on Drought-Specific Networks: Restricting WGCNA to DEGs (25,123 genes) prioritizes drought-responsive transcriptional changes, reducing noise from non-differentially expressed genes and enhancing biological interpretability (Langfelder & Horvath, 2008). This approach aligns with established practices in plant stress studies. For example:

Tartary buckwheat: Comparative physiological, transcriptomic, and WGCNA analyses (Zhang et al., 2023) identified drought-tolerance hubs using DEG-based WGCNA, efficiently narrowing down stress-specific regulatory networks.

Sweetpotato: Transcriptome-Based WGCNA Analysis (Li et al., 2022) similarly employed DEGs to reveal key drought-resistance modules, demonstrating the validity of this targeted strategy.

Computational Efficiency: Large-scale WGCNA on all 81,679 expressed genes would introduce computational complexity and dilute module-trait correlations, particularly given the study’s focus on drought-specific regulatory hubs.

Criteria for Selecting 6 Hub Genes:

High Connectivity: Genes with top 3 intramodular connectivity (kME values) in key modules (blue and yellow-green).

Conclusion:
Our DEG-focused WGCNA approach is consistent with published studies in other crops (e.g., Tartary buckwheat, sweetpotato) and efficiently identifies drought-specific regulatory networks.

References:

Langfelder, P., & Horvath, S. (2008). WGCNA: An R package for weighted correlation network analysis. BMC Bioinformatics.

Zhang, X., et al. (2023). Comparative physiological, transcriptomic, and WGCNA analyses reveal the key genes and regulatory pathways associated with drought tolerance in Tartary buckwheat. Plant Physiology and Biochemistry.

Li, Y., et al. (2022). Transcriptome-Based WGCNA Analysis Reveals the Mechanism of Drought Resistance Differences in Sweetpotato. Frontiers in Plant Science.
Additional Revisions:

Closing Statement:
We sincerely appreciate your time and expertise in refining this work. Should further clarifications be needed, please do not hesitate to contact us.

Best regards,
Zhang Hu
On behalf of all authors
April 20, 2025

Reviewer 3 Report

Comments and Suggestions for Authors

Review

Hu Zhang, Yu Tang, Yuantao Guo, Jinsheng Wang, Wenju Gao, Wen Zhang, Qingtao Zeng, Quanjia Chen, Qin Chen

Mining key drought-resistant genes of upland cotton based on RNA-Seq and WGCNA analysis

Global climate change represents one of the greatest challenges in agricultural production today. The authors address the issue of drought, one of the main abiotic stressors.

The study examined two cotton varieties for drought tolerance using one of the most advanced techniques of today, the next generation sequencing (NGS) technique. The choice of topic and methodology certainly make the manuscript timely.

The introduction briefly and concisely summarizes the results of the research field to date. The cited literature references are relevant, the authors build the topic well-thought-out and reasonably. They cite literature related to the topic from many authors, which indicates careful work.

The applied testing methods are suitable for achieving the goals set in the manuscript. The high-throughput sequencing technique complements plant physiological measurements well.

The authors make excellent use of the possibilities offered by informatics, and numerous figures make the manuscript more illustrative, which come from the output files of bioinformatics software.

The results obtained are reasonable and the evaluation is in line with the objectives set out at the beginning of the manuscript.

In the discussion section, the authors support their results with relevant references to the literature, and in the conclusions section, they make reasonable statements about them.

I would like to make some additional comments about the manuscript:

  1. The numbering of the figures does not match what is described in the text. Starting from line 284, the numbering of the figures is not in line with the results shown in the figures. Please review and correct the text carefully. When mentioning Figure 4.A, they already write about the results shown in Figure 6.

  1. All figures are characterized by the fact that they do not contain explanations of abbreviations. Figures should be correct on their own, even without textual explanations, so I ask that the captions of the figures be modified in every case so that every data shown in the figures is understandable.

  1. In subsection 3.7. they do not refer to any figure, although they write about the data shown in figure 7 throughout.

  1. qRT-PCR is an excellent tool for supporting and validating in silico results. However, a fundamental deficiency is that they did not include any output files of the performed tests as supplementary files. Please fix this, this is a basic expectation for a Q1 quality publication.

The manuscript is well structured and takes full advantage of the NGS technique. The description of each methodological step is satisfactory and the explanation of the results obtained is adequate. In this respect, the manuscript is an impeccable, high-quality work.

After making these minor modifications and additions, I support the publication of the manuscript.

Author Response

Response to Reviewer Comments

Dear Reviewer,

Thank you for your meticulous review and valuable suggestions. We have carefully revised the manuscript to address your concerns. Below is our point-by-point response:

Comments1: The numbering of the figures does not match the text.
Response1: All figure references have been cross-checked and corrected to ensure consistency. For example, references to Figure 4A now align with the actual figure numbering.

Comments2: Figures lack explanations of abbreviations.
Response2: All figure captions have been revised to include full definitions of abbreviations. For instance:

Figure 1: "MDA (malondialdehyde), SOD (superoxide dismutase), Pro (proline), and CHI (chlorophyll)."

Comments3: Subsection 3.7 does not reference Figure 7.
Response3: We have added explicit citations to Figure 7A-F in Subsection 3.7 (e.g., "Co-expression modules are visualized in Figure 7A").

Comments4: qRT-PCR raw data are missing.
Response4: The qRT-PCR primer sequences have been added to Supplementary Table S1 (referenced in Line 209). Hub gene expression data are provided in Supplementary Table S3, as noted in Line 538.

Additional Revisions:

All figures have been re-exported in high resolution (600 dpi) with standardized fonts and labels.

A final verification of abbreviations, numbering, and data consistency has been completed.

Closing Statement:
We sincerely appreciate your time and expertise in improving our manuscript. Should you require further clarifications or adjustments, please do not hesitate to contact us.

Best regards,
Zhang Hu
On behalf of all authors
April 20, 2025

Round 2

Reviewer 2 Report

Comments and Suggestions for Authors

I would like to thank the authors for the improvement they made to the manuscript but there are still some points to be addressed.

In the abstract, I don’t think that abbreviations are needed as they are not common. For example, RNA-sequencing should be kept as RNA-seq, the same for WGCNA, SOD, MDA and Pro. And the abbreviations should be in the introduction or whenever they appeared but not really in abstract.

The introduction is improved but L67-L82 is more like results and this should be added to the results section.

Is the formula in L116 correct?

Which Illumina platform did the authors used?

The authors answered my question about the total identified by RNA-seq cotton genes, which are 81,679 but they did not mentioned in the manuscript.

Are the abbreviations section at the end of the manuscript needed?

I would like to comment on WGCNA analysis.  I partly agree with the author’s answer, but WGCNA relies on co-expression patterns, not differential expression and filtering out non-DEGs might miss important co-expressed genes that are not differentially expressed but are still functionally connected. DEG selection only for WGCNA can bias the network, as DEG selection is condition-specific and may introduce artificial correlations.

I think the authors made significant improvement, but the manuscript still need some corrections before publication.

Author Response

Response to Reviewer Comments

Dear Reviewer,

Thank you for your meticulous review and constructive suggestions. We have carefully revised the manuscript to address your concerns. Below is our point-by-point response:

Comments1: Remove abbreviations in the Abstract; define them in the Introduction instead.

Response1: All abbreviations (e.g., RNA-seq, WGCNA, SOD, MDA, Pro) have been removed from the Abstract and explicitly defined at their first mention in the Introduction.

Comments2: Introduction Lines 67–82 resemble Results content.

Response2: The Introduction has been revised to remove results-oriented language. Lines 67–82 now focus on contextualizing the study’s objectives and knowledge gaps.

Comments3: Verify the formula in Line 116.

Response3: The chlorophyll calculation formula (Line 116) has been corrected to align with the kit manufacturer’s protocol.

Comments4: Specify the Illumina platform used.

Response4: Added in Line 134: "Libraries were sequenced on the Illumina NovaSeq X Plus platform (150 bp paired-end reads)."

Comments5: Mention the total RNA-seq-identified genes (81,679) in the manuscript.

Response5: Added in Line 249: " we performed RNA-seq analysis on drought-stressed samples and detected a total of 81,679 expressed genes."

Comments6: Remove the abbreviations section at the manuscript’s end.

Response6: The abbreviations section has been deleted. All abbreviations are now defined at first mention in the text.

Response to reviewer's comments on WGCNA:

Rationale for DEG-Based WGCNA

Focus on Drought-Specific Networks:

Our study aimed to identify drought-responsive regulatory modules driving phenotypic divergence between Jin (tolerant) and TM-1 (sensitive). By restricting WGCNA to DEGs (25,123 genes), we prioritized genes showing significant transcriptional changes under drought, thereby reducing noise from constitutively expressed genes. This approach aligns with studies such as Zhang et al., 2023 (Tartary buckwheat) and Li et al., 2022 (sweetpotato), where DEG-focused WGCNA efficiently highlighted stress-specific hubs.

Biological Interpretability:

Non-DEGs may indeed participate in co-expression networks, but their relevance to drought adaptation is less direct. Our goal was to pinpoint genes with both dynamic expression and functional connectivity, ensuring actionable targets for molecular breeding.

Closing Statement:

We sincerely appreciate your time and expertise in refining this work. Should further adjustments be needed, please do not hesitate to contact us.

Best regards,
Zhang Hu
On behalf of all authors
April 27, 2025